# Efficacy of the PRESERFLO MicroShunt and a Meta-Analysis of the Literature

**DOI:** 10.3390/jcm11237149

**Published:** 2022-12-01

**Authors:** Shigeo S. M. Pawiroredjo, Wichor M. Bramer, Noemi D. Pawiroredjo, Jan Pals, Huub J. Poelman, Victor A. de Vries, Roger C. W. Wolfs, Wishal D. Ramdas

**Affiliations:** 1Department of Ophthalmology, Erasmus Medical Center, University Medical Center, 3000 CA Rotterdam, The Netherlands; 2Medical Library, Erasmus Medical Center, University Medical Center, 3000 CA Rotterdam, The Netherlands; 3Faculty of Science, Vrije Universiteit Amsterdam, 1081 HV Amsterdam, The Netherlands

**Keywords:** PRESERFLO, MicroShunt SIBS polymer, ab externo surgery, MIGS, glaucoma, intraocular pressure (IOP)

## Abstract

Background: Recent studies on the PRESERFLO MicroShunt suggest that it may be effective in lowering intraocular pressure (IOP); however, the number of studies on this device remains limited. Therefore, we assessed the efficacy of the PRESERFLO MicroShunt in patients with glaucoma and performed a meta-analysis of published results. Methods: Prospective study including all patients that underwent PRESERFLO MicroShunt surgery from 2018 onwards. Sub-analyses were performed for cataract-combined procedures. To compare our results, we performed a systematic review and meta-analysis. IOP, IOP-lowering medication and surgical complications reported in the retrieved studies were assessed. Results: A total of 72 eyes underwent PRESERFLO-implant surgery (59 as standalone procedure and 13 as cataract-combined procedure). No significant differences were found in IOP and IOP-lowering medication between both groups. The mean ± standard deviation IOP and IOP-lowering medications of both groups taken together declined from 21.72 ± 8.35 to 15.92 ± 8.54 mmHg (*p* < 0.001, 26.7% reduction) and 3.40 to 0.93 (*p* < 0.001, 72.6% reduction) at 1 year follow-up, respectively. Secondary surgeries were required in 19.4% of eyes, the majority (71.4%) within 6 months. The meta-analysis including 14 studies (totaling 1213 PRESERFLO MicroShunt surgeries) from the systematic review showed a mean preoperative IOP and IOP-lowering medication of 22.28 ± 5.38 and 2.97 ± 1.07, respectively. The three-years postoperative pooled mean was (weighted mean difference, 95% CI) 11.07 (10.27 [8.23–12.32], *p* < 0.001) mmHg and 0.91 (1.77 [1.26–2.28], *p* < 0.001) for IOP and IOP-lowering medication, respectively. The most common reported complication was hypotony (2–39%). Conclusion: The PRESERFLO MicroShunt is effective and safe in lowering IOP and the number of IOP-lowering medications.

## 1. Introduction

Glaucoma is an optic neuropathy which is often caused by a rapid increase and/or prolonged periods of elevated intraocular pressure (IOP) [1]. Treatment of glaucoma is mainly focused on lowering of the IOP [2]. The first-line treatment of glaucoma usually consists of IOP-lowering medication [3]. If the target IOP is not achieved using IOP-lowering medication, laser treatment or incisional surgery is introduced [1]. Trabeculectomy is generally regarded as the gold standard among surgical glaucoma treatments for many years and has shown to drastically reduce IOP; however, postoperative interventions are frequently required because of the high risk of complications [4,5]. Therefore, there has been an increased demand for safer and less invasive surgical treatment options. These are the so-called minimally invasive glaucoma surgeries (MIGS) and aim to reduce the IOP whilst lowering the risk of complications [6,7]. One of these new implants is the PRESERFLO MicroShunt (Santen Inc., Miami, FL; formerly known as the InnFocus MicroShunt). The PRESERFLO MicroShunt is an 8.5 mm long tube with a 70 µm lumen composed of poly(styrene-block-isobutylene-block-styrene), or SIBS. This device is implanted via an ab externo approach in the anterior chamber-angle, allowing outflow of aqueous humor from the anterior chamber to a posterior subconjunctival bleb [8]. Implantation of the PRESERFLO MicroShunt could be performed standalone or in combination with cataract surgery and is augmented with mitomycin-C to prevent postoperative scarring [9,10]. Recent studies on the PRESERFLO MicroShunt suggest that it is effective in lowering IOP and the number of IOP-lowering medications; however, the number of studies on this device remains limited. The aim of this study is to assess the efficacy and safety of the PRESERFLO MicroShunt as a standalone and a cataract-combined procedure in patients with glaucoma. Furthermore, a systematic review with meta-analysis was conducted to see whether the results of the current study are in line with the available literature and to give an overall view of its performance across multiple studies.

## 2. Materials and Methods

### 2.1. Study Design

The current study was conducted in a subset of the Erasmus Glaucoma Cohort [11]. For the current analyses, a prospective single-center study was conducted. All patients who underwent PRESERFLO MicroShunt implantation at the department of ophthalmology of the Erasmus Medical Center, Rotterdam, The Netherlands, from July 2019 till June 2021 were included in the current study.

All patients underwent extensive ophthalmologic examination including best corrected visual acuity, autorefraction, gonioscopy, and applanation tonometry. Data (IOP and number of IOP-lowering medications) on 1 day, 1 week, 1 month, 3 months, 6 months, 1 year, and the last postoperative visit were collected. The Medical Ethics Committee of the Erasmus University had approved the study. Formal consent was not required, because patients did not undergo non-clinically related interventions.

### 2.2. Surgical Procedure

All surgeries were either performed by one of the two surgeons (RCWW and WDR). First, local or general anesthesia was administered. A 6 to 8 mm incision was made through the conjunctiva at the limbus and the Tenon’s capsule was dissected from the sclera using blunt tipped scissors. Three LASIK sponges with 0.2 mg/mL mitomycin-C were then placed beneath the conjunctiva flap for 3 min. The flap was then washed with a sterile saline solution. Bipolar diathermy was used to avoid bleeding (if required). An inked marker was then used to mark a point on the sclera 3 mm posterior from the limbus. Next, a shallow 2 mm scleral pocket was formed using a 1 mm slit knife. A needle tract was made into the anterior chamber using a 25-gauge needle through the scleral pocket, bisecting the iridocorneal angle. The MicroShunt was then inserted through the needle tract using forceps with the bevel up, until the wedge fins of the MicroShunt locked into the scleral pocket to avoid migration. Flow of aqueous humor was confirmed by priming and observing drop formation at the distal end of the MicroShunt. Once the flow was confirmed, the distal end of the MicroShunt was tucked under the Tenon’s capsule and the Tenon’s capsule was then pulled up and over the MicroShunt. Lastly, the conjunctiva was closed using Vicryl 8-0 sutures.

The main indication for cataract surgery was a narrow anterior chamber or significant cataract. In case of a combined procedure, cataract surgery was performed directly after administrating mitomycin-C and before the needle tract was created.

A secondary surgery was indicated if the MicroShunt failed to achieve the desired reduction in IOP. Secondary surgery was performed as follows: the conjunctiva of the bleb area was opened, and all the tissue adhesions were removed. Mitomycin-C was not used. Next, the PRESERFLO MicroShunt was primed. Lastly, the conjunctiva was closed with Vicryl 8-0 sutures.

### 2.3. Assessment of Main Outcomes

IOP was assessed using Goldman applanation tonometry (Haag-Streit, Köniz, Switzerland), which had been calibrated according to manufacturer’s recommendations. IOP-lowering medications were divided into several categories which included: prostaglandins, beta-blockers, carbonic anhydrase inhibitors, oral acetazolamide, and alfa-2 agonists. Fixed combinations of eye drops were calculated as two separate drugs. The number of IOP-lowering medications was then calculated by adding the number of categories to one another [12]. Hypotony was defined as an IOP ≤ 5 mmHg at two or more consecutive postoperative visits (excluding one day postoperative).

### 2.4. Search Strategy, Study Eligibility and Quality Assessment

For the second part of the study, a systematic review and meta-analysis of the literature was conducted in which the Embase, Medline ALL Ovid (Pubmed), Web of science (SCI-EXPANDED & SSCI, 1975) and Cochrane CENTRAL register of Trials were searched up to 24 May 2022 (date last searched). Reporting of the systematic review was achieved by adhering to the Preferred Reporting Items for Systematic review and Meta-Analyses (PRISMA) and the Meta-analysis Of Observational Studies in Epidemiology (MOOSE) guidelines [13,14]. Firstly, studies with an available abstract, with human participants, and studies of which a full text was available in English were independently screened by two researchers (SSMP and NDP). Secondly, the full text was read to assess whether an article was eligible for inclusion and the reference list was scanned to find additional eligible studies. Finally, the results were compared, and discrepancies were discussed. If the two researchers could not reach consensus, a third researcher (WDR) was involved to clarify. Studies had to report on pre- and post-operative IOP and/or IOP-lowering medication. Exclusion criteria were case reports or a follow-up of less than 6 months. If the same study population was used in multiple studies, only one study was included. Next, relevant data was extracted from the articles which included author, sample size, publication year, diagnosis, study design, follow-up, number of secondary surgeries, occurrence of complications, pre- and post-operative IOP and IOP-lowering medications, and proportion of patients without any IOP-lowering medication at follow-up. To assess the methodological quality of the individual studies, the Newcastle–Ottawa Scale for assessing the quality of comparative non-randomized studies was used [15].

### 2.5. Statistical Analysis

Baseline characteristics were analyzed using the independent t-test for continuous data and the chi-square test (or Fisher’s exact test if applicable) for categorical data. The paired samples t-test was used for within subgroup analysis (e.g., pre- and post-operative IOP).

Differences in IOP and IOP-lowering medication over time were assessed using linear mixed models. Two models were created both assuming an unstructured correlation matrix, in which one of the variables was fitted as the dependent variable with visit (fixed) as a factor. The models were adjusted for age and gender (both fixed) and accounted for using both eyes from the same individual. Kaplan–Meier analyses were performed in which failure was defined as requiring secondary surgery (i.e., bleb revision or placement of another glaucoma drainage device). Follow-up was counted as the date of first surgery until the date of requiring secondary surgery. If an eye did not require secondary surgery, follow-up was counted as the date of first surgery until the date of the last postoperative visit. A separate Kaplan–Meier analysis was performed in which failure was defined according to the criteria by the World Glaucoma Association (WGA) [16]. Surgery was considered a failure if a patient had <20% reduction in IOP from baseline or if the IOP was out of target range (5–18 mmHg, inclusive) for two consecutive visits. Both counted after the first month postoperative onwards. If an eye met the failure criteria as defined by the WGA, follow-up was counted as the day of surgery until the first day failure was noted. The log-rank test was used to assess statistically significant differences between PRESERFLO as a standalone procedure and PRESERFLO as a cataract-combined procedure.

Statistical analyses were performed using SPSS v.25 for Windows (SPSS Inc., Chicago, IL, USA). Statistical significance was considered if *p* < 0.05.

Meta-analyses were performed using Revman (RevMan 5.3 for Windows and Mac; The Cochrane Collaboration, Oxford, UK). The mean and standard deviations were extracted to calculate the weighted mean difference (WMD) with corresponding 95% CI and the pooled mean with standard deviation for the specified time intervals. For the meta-analyses, fixed-effect models were used to pool the results. I2 statistics were calculated to assess heterogeneity.

## 3. Results

A total of 72 eyes (of 60 patients) underwent PRESERFLO MicroShunt implantation of whom 59 underwent a standalone procedure. The median (interquartile range) follow-up was 0.72 (0.33–1.12) and 1.15 (0.65–1.66) years for the standalone procedure and the cataract-combined procedure, respectively. One patient in the standalone procedure group was lost to follow-up at 1 month. Patients who underwent a standalone procedure had a significantly higher central corneal thickness compared to patients who underwent a cataract-combined procedure (*p* = 0.025; Table 1).

The IOP-levels and number of IOP-lowering medications for the PRESERFLO MicroShunt as a standalone procedure and as a cataract-combined procedure are presented in Figure 1A. The differences in IOP for PRESERFLO MicroShunt as a standalone procedure and as a cataract-combined procedure are graphically presented in Appendix A. The mean reduction in preoperative IOP and IOP-lowering medication for the standalone and combined procedures were similar 5.07 mmHg (23.6%) and 8.92 mmHg (38.2%; *p* = 0.234), and 2.67 (75.5%) and 1.62 (56.8%; *p* = 0.830), respectively, at the last postoperative visit. Moreover, the linear mixed model showed no significant differences in IOP and IOP-lowering medication if the whole follow-up period was taken into account (*p* = 0.176 and *p* = 0.548, respectively). Figure 2A shows a Kaplan–Meier curve in which cumulative failure is defined as requiring secondary surgery. Secondary surgeries were required in 12 eyes (20.3%) with PRESERFLO MicroShunt as a standalone procedure and in 2 eyes (15.4%) with PRESERFLO as a cataract-combined procedure, respectively (*p* = 1.000). Of the 14 (19.4%) eyes that required secondary surgery, 10 (71.4%) underwent surgery within 6 months after primary surgery. Figure 2B shows the cumulative failure rate as defined by the WGA-criteria. Of all operated eyes, 66.7% did not require any IOP-lowering medication at the last follow-up. Hypotony was observed in 3 eyes (5.1%) with PRESERFLO MicroShunt as a standalone procedure and in 2 eyes (15.4%) with PRESERFLO as a cataract-combined procedure (*p* = 0.219). All hypotony cases resolved spontaneously. There were no significant differences in visual acuity pre- and post-operative within/between both groups. We observed no serious complications (i.e., endophthalmitis or blebitis).

As there were no significant differences between the standalone and the cataract-combined procedure, we merged both groups (*n* = 72) for further analysis. Figure 1B shows the IOP levels and IOP-lowering medication for the total study population and separately for the eyes that underwent secondary surgery (*n* = 14 of 72 eyes). Eyes that required secondary surgery had a similar preoperative IOP-level but a significantly lower number of IOP-lowering medications (mean ± standard deviation: 26.17 ± 11.73 mmHg and 21.72 ± 8.35 mmHg (*p* = 0.67) and 3.40 ± 1.37 vs. 1.61 ± 1.54 (*p* < 0.001), respectively) compared to eyes that did not require secondary surgery.

The literature search yielded a total of 311 articles of which 14 studies (including 1213 PRESERFLO MicroShunt surgeries) were included in the current study and considered eligible for the meta-analysis (Appendix A) [9,17,18,19,20,21,22,23,24,25,26,27,28,29,30]. Follow-up ranged from 12 months to 72 months. The median quality score (range) according to the Newcastle–Ottawa scale was 8 (7–8) on a scale from 0–9 (Appendix A). Most studies included a mixture of different types of glaucoma with a majority having primary open-angle glaucoma (similar to our dataset). Table 2 shows the prevalence of complications across all included studies. The most common complications were hypotony (1.7–39%) and choroidal effusion/detachment (2.0–12.9%). Figure 3A,B show a summary of the meta-analysis for the performance of the PRESERFLO MicroShunt on the IOP and the IOP-lowering medication for all specified time intervals (for the full meta-analyses see Appendix A).

After 3 years of follow-up, the WMD (95%, CI; I2) IOP and IOP-lowering medication was 10.27 mmHg (8.23–12.32; 91%) and 1.77 (1.26–2.28; 0%), respectively. However, as only a few studies had such a long follow-up and to make the results more comparable to our data, we also report the 1 year WMD (95% CI; I2) for the IOP and IOP-lowering medication: 9.04 mmHg (8.46–9.63; 88%) and 2.67 (2.65–2.70; 89%) at 1 year, respectively.

The mean ± standard deviation preoperative IOP and IOP-lowering medication were 22.28 ± 5.38 mmHg and 2.97 ± 1.07, which declined to 14.09 ± 4.09 mmHg and 0.63 ± 1.00 at 1 year follow-up, respectively. Approximately 57% of eyes were free of any IOP-lowering medication after MicroShunt implantation. In our study population, eyes that underwent PRESERFLO MicroShunt surgery had a mean difference (95% CI) of 5.75 (3.19–8.30) mmHg and 2.47 (2.00–2.93) in IOP and IOP-lowering medication at the last postoperative visit, respectively.

## 4. Discussion

The PRESERFLO MicroShunt was found to significantly reduce IOP and number of IOP-lowering medications with 26.5% and 72.7%, respectively. At the end of follow-up, 66.7% of eyes did not require any IOP-lowering medication. As far as we know, the current study presents the first meta-analysis on the efficacy of the PRESERFLO MicroShunt. The total sample size included over one thousand PRESERFLO MicroShunt surgeries.

The preoperative IOP in the meta-analysis was higher than in the current study (22.28 vs. 21.70 mmHg). The preoperative number of medications were lower in the meta-analysis than in the current study (2.96 vs. 3.40). This may explain the finding that the absolute reduction in IOP was greater in the meta-analysis (11.21 mmHg vs. 5.75 mmHg) while the reduction in IOP-lowering medication was larger in our study population (2.47 vs. 2.06). The meta-analysis showed a WMD (95% CI; I2) for IOP of 9.04 mmHg (8.46–9.63; 88%) at 1 year, which is greater than the mean difference found in our study (5.56 mmHg [3.12–7.99]). The higher WMD in the meta-analysis is caused by a high WMD in four individual studies [18,20,21,22]. Additionally, the meta-analysis showed an interesting finding, namely that there was a decrease in IOP whilst also a slight increase of IOP-lowering medication use at 3 years postoperative. A possible explanation for this finding could be that only two studies reported on IOP and IOP-lowering medication at 3 years follow-up, of which one study reported a decrease in IOP whilst also reporting an increase in the number of IOP-lowering medications [18]. However, the authors of the study did not further discuss this finding.

The proportion of eyes requiring secondary surgery related to the PRESERFLO MicroShunt in our study population (19.4%) was higher than other studies reporting 1.2–15.6% (Table 2). It is important to note that studies used different surgical approaches when performing secondary surgery; some dissected the conjunctiva to remove overlying scar tissue with the Tenon’s capsule and the PRESERFLO MicroShunt was replaced if needed, while others used needling with or without 5-fluorouracil (5FU) behind the slit lamp. An interesting finding was that if a patient required a secondary surgery, the surgery was usually performed within 6 months (71.4%; Figure 2A), which is comparable to the XEN-implants (63% within 6 months), another MIGS procedure [11]. Hypotony observed in our study (5 of 72 eyes (6.7%)) was in line with the literature; however, it is important to note that the reported range was wide (1.7–39%). There was an increase in number of IOP-lowering medications up to the “last” time interval (with a median [interquartile range] follow-up of 1.02 [0.68–1.46]); however, only the increases between 3 and 6 months and 6 months and 1 year were statistically significant. Similar to our study population, the meta-analysis also found a slight increase in the number of IOP-lowering medications at 2–3 years postoperative (*p* = 0.41).

A previously published study comparing trabeculectomy to the PRESERFLO MicroShunt found the MicroShunt to be inferior to trabeculectomy in regard to the proportion of eyes that met the criteria for surgical success (53.9% vs. 72.7%, respectively) [28]. Their study confirmed the favorable safety profile of the PRESERFLO MicroShunt, with patients in the MicroShunt group having a significantly lower incidence of hypotony compared to patients that underwent trabeculectomy (28.9% vs. 49.6%, respectively). Furthermore, fewer patients in the MicroShunt group required postoperative interventions compared to the trabeculectomy group (40.8% vs. 67.8%, respectively) [28]. Another study found that the XEN-implant did not show any significant differences between XEN-implant with or without a cataract-combined procedure for failure as requiring secondary surgery [11]. Similarly to its equivalent, we found that the PRESERFLO MicroShunt did not show significant differences between PRESERFLO MicroShunt as a standalone procedure and as a cataract-combined procedure for cumulative failure, defined as requiring secondary surgery (Figure 2A; log-rank *p* = 0.524) [11]. Nonetheless, it should be noted that it is difficult to compare the results of different (non-comparative) studies, because of different populations, surgery indications, and outcome criteria.

Our study has several limitations. Firstly, the surgeon may have a certain preference for a specific procedure in a particular situation (e.g., difference between standalone and combined procedure), which may result in selection bias. Secondly, the results of the cataract-combined procedures should be taken with caution, as the size of this subgroup was low and the indication to combine the surgery with cataract-extraction might be driven by the status of the anterior chamber angle. Thirdly, surgery was performed by two surgeons, which could contribute to variability in outcomes. However, no significant differences in outcomes were found for postoperative IOP and number of IOP-lowering medications between both surgeons. Moreover, the rates of secondary surgery between both surgeons were similar 16.1% and 22.0% (*p* = 0.537).

The arising question is: Will the PRESERFLO MicroShunt be able to replace trabeculectomy and glaucoma drainage devices (GDDs) in the near future? The answer is probably “no”. The IOP decreasing efficacy seems less than, for example, the Ahmed or Baerveldt implant or a trabeculectomy. A recent study showed a 39.1% and 38% IOP reduction for the Baerveldt implant and trabeculectomy, respectively [31]. The safety profile of the PRESERFLO MicroShunt in terms of complications, however, seems better than traditional GDDs [32]. Therefore, MIGS might be a good first choice if a surgical intervention is required to lower IOP.

## 5. Conclusions

The PRESERFLO MicroShunt is effective in lowering IOP in patients with glaucoma. The IOP-lowering performance seems less than traditional surgical procedures; however, the MicroShunt seems to have a better risk profile than conventional glaucoma surgeries. Future studies in the form of randomized controlled clinical trials with long follow-ups are needed to confirm the current results and to establish the place of the PRESERFLO MicroShunt within glaucoma treatment paradigm.

## Figures and Tables

**Figure 1 jcm-11-07149-f001:**
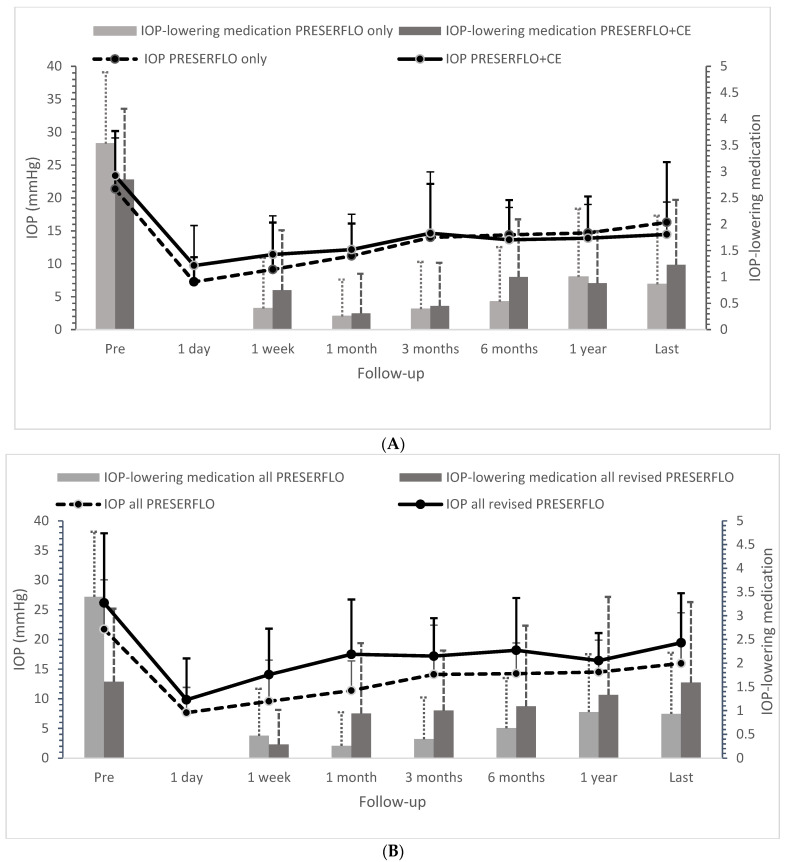
(**A**,**B**) Mean intraocular pressure (IOP, lines) and mean number of IOP-lowering medication (bars) for PRESERFLO as a standalone procedure vs. as a cataract-combined procedure (**A**) and all Preserflo surgeries (including combined procedures) vs. secondary surgeries (all revised Preserflo implants) (**B**). The “last” time interval had a median (interquartile range) follow-up of 1.02 (0.68–1.46) years. The vertical lines represent the standard deviation (SD). IOP = intraocular pressure; CE = cataract extraction.

**Figure 2 jcm-11-07149-f002:**
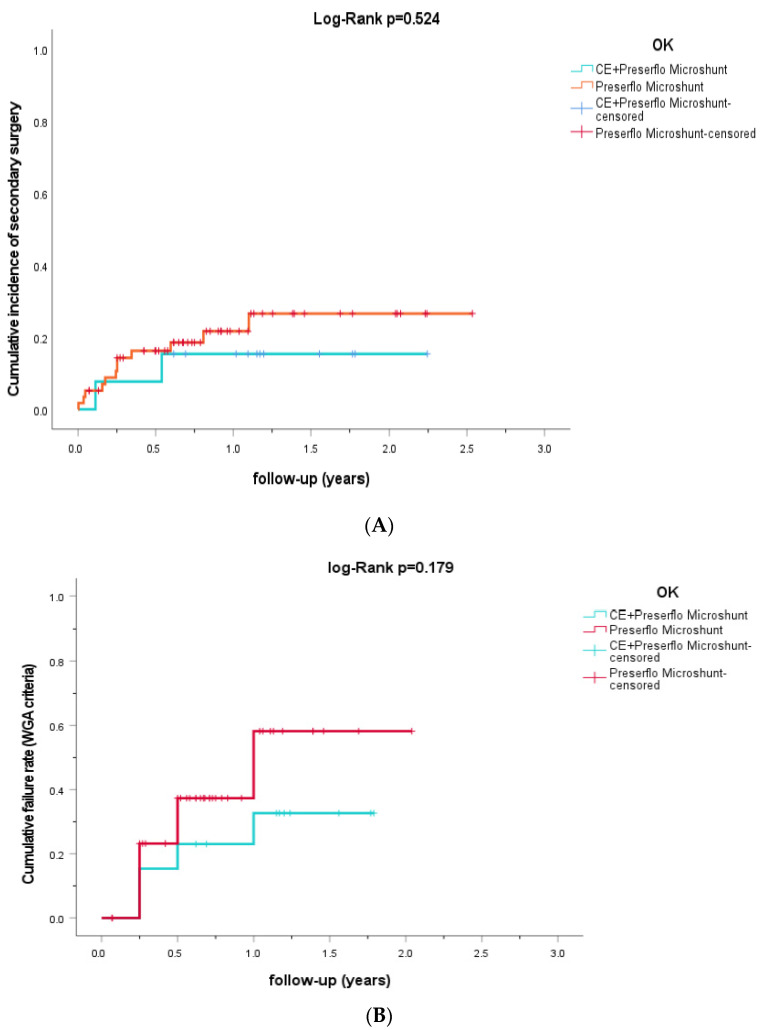
(**A**,**B**) Kaplan–Meier cumulative incidence curve for failure defined as requiring secondary surgery (**A**) and according to the WGA criteria (**B**) for PRESERFLO MicroShunt as a standalone procedure and as a cataract-combined procedure. Censored patients are represented by vertical tick marks. CE = cataractextraction, WGA = World Glaucoma Association.

**Figure 3 jcm-11-07149-f003:**
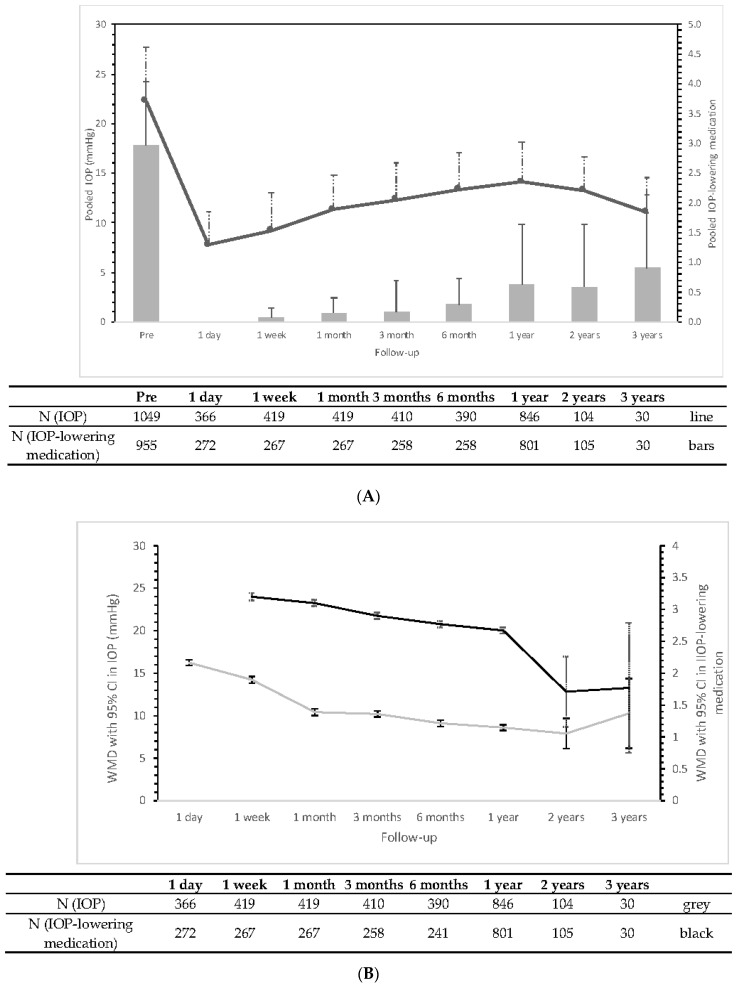
(**A**,**B**) Meta-analyses for the change in intraocular pressure (IOP; grey) and number of IOP-lowering medications (black) after PRESERFLO MicroShunt surgery. Presented as the pooled mean with standard deviation (**A**) and weighted mean difference (WMD) with corresponding 95% confidence intervals (**B**). Table below gives the number of eyes.

**Table 1 jcm-11-07149-t001:** Baseline preoperative characteristics of the population presented as the mean ± standard deviation unless stated otherwise.

	PRESERFLO MicroShunt (*n* = 59 Eyes)	PRESERFLO MicroShunt Combined with Cataract Extraction (*n* = 13 Eyes)	*p*-Value
Age (years)	67.11 ± 10.4	70.32 ± 13.4	0.343
Gender, female (*n*,%)	24 (40.6)	3 (23)	0.346
Caucasian descent (*n*,%)	30 (50.8)	7 (53.8)	0.384
Untreated IOP at diagnosis (mmHg)	25.0 ± 10.7	24.9 ± 8.5	0.957
IOP (mmHg)	21.4 ± 8.8	23.4 ± 5.8	0.311
Number of IOP-lowering medication	3.5 ± 1.4	2.9 ± 1.4	0.106
Visual acuity	0.75 ± 0.41	0.75 ± 0.23	0.942
Central corneal thickness (µm)	542.3 ± 36.3	514.3 ± 45	0.025
Previous intraocular surgery (*n*,%) *	14 (23.7)	0 (0)	0.059
Follow-up (median [IQR])	0.72 (0.33–1.12)	1.15 (0.65–1.66)	0.111

* = cataract surgery (24 of 59 eyes) not counted; IOP = intraocular pressure; IQR = interquartile range.

**Table 2 jcm-11-07149-t002:** Prevalence of complications according to the systematic review.

Complications	Median (%)	Range(%)
Incorrect location/positioning	1.9	1.0–8.6
Choroidal effusion/detachment	8.9	2.0–12.9
Corneal dellen	1.2	1–1.2
Corneal edema	1.2	1–1.2
Hyphaema	6.8	2.5–22.7
Hypotony	11.1	1.7–39
Hypotonic maculopathy	1.1	0.8–6.9
Implant blocking/fracture/migration	2	1–4.3
Macula edema	1	0.6–3.6
Ptosis	2	1.2–2.4
Shallow/flat anterior chamber	5.7	2.5–13
Wound leak/seidel	4.9	0.6–8.9
Vitrious hemorrhage	1.2	0.6–4.3
Needling	12.2	1.6–62.5
Surgical revision	6.5	1.2–15.6

## Data Availability

The data presented in this study are available on request from the corresponding author.

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
