# Peer review of "Efficacy of the PRESERFLO MicroShunt and a Meta-Analysis of the Literature"

_jcm, 2022, doi:10.3390/jcm11237149_

Round 1

Reviewer 1 Report

Pawiroredjo et al. present a well-written manuscript analyzing the IOP lowering effect, the reduction of medications and the complication profile of Preserflo implantation in a clinical Rotterdam cohort, as well as in a systematic review with meta-analysis. The reported methods are adequate and consistent. 

Some minor comments:

Please further explain: „Positive medical history“ in table 1.

Please report in Table 1 the number of eyes with previous cataract surgery in the group without cataract surgery.

Table 2: needling and surgical revision do not reflect a complication but rather a consequence due to complications. Thus, I would somehow add these items at the end of the list and visualize them differently.

Figure 3 shows the IOP over time after Preserflo implantation. I recommend to delete the data for 3 years as they only rely on 30 eyes (from one study?) compared to several hundreds at the other time points.

Results: the authors should further report how many included studies were RCTs, prospective observational studies and retrospective studies and whether the results did differ between these different studies.

Discussion: the authors might further discuss that Preseflo implantation (RCT: Reference 28) did show a similar IOP lowering effect in those eyes not having normal-tension glaucoma (non-inferiority), while not in those eyes with normal-tension glaucoma.

Author Response

Reviewer: 1

Comments to the Author

Pawiroredjo et al. present a well-written manuscript analyzing the IOP lowering effect, the reduction of medications and the complication profile of Preserflo implantation in a clinical Rotterdam cohort, as well as in a systematic review with meta-analysis. The reported methods are adequate and consistent.

Some minor comments:

Please further explain: „Positive medical history“ in table 1.

REPLY and Change: This number reflected the number of comorbidities; however, we agree with the reviewer that it does not make sense to mention these numbere. We removed this row from the table.

Please report in Table 1 the number of eyes with previous cataract surgery in the group without cataract surgery.

REPLY and Change: We have added this number with percentage to the footnote of Table 1.

Table 2: needling and surgical revision do not reflect a complication but rather a consequence due to complications. Thus, I would somehow add these items at the end of the list and visualize them differently.

REPLY and Change: We have moved both to a lower/separate position in the Table 2. Furthermore, we changed the order to an alphabetical order and corrected a typo.

Figure 3 shows the IOP over time after Preserflo implantation. I recommend to delete the data for 3 years as they only rely on 30 eyes (from one study?) compared to several hundreds at the other time points.

REPLY: We agree that the last time point includes “only 30” eyes. This is due to the fact that there are not many studies with such a long follow-up. In the current meta-analysis this time point includes two studies. Moreover, in our own data the eye with longest follow-up had a follow-up of over 2.5year with results that are comparable to that of the meta-analysis.

Results: the authors should further report how many included studies were RCTs, prospective observational studies and retrospective studies and whether the results did differ between these different studies.

REPLY and Change: Table S1 shows this information. We noticed that there was no reference to this table. This has now been added to the Results section.

Reviewer 2 Report

Dear authors,

Congratulations on successfully conducting this relevant and well-designed study. I only have a few comments/ questions about your paper.

Major comments:

Could you make a statement about the glaucoma etiology of the patients included in the study (primary/ secondary glaucoma, exfoliation, etc.), as this is important to get an idea about the severity/ aggressivity of the disease.

Regarding the meta-analysis, would it be possible to also analyze PRESERFLO with vs. without combined phacoemulsification?

Minor comments:

Methods:

It seems like only one eye per patient was analyzed, but it is not clearly stated in the methods section. It would be important to clarify this.

Is it true, that you didn’t use any antimetabolite for the bleb revisions?

Results:

Line 106: typo in prostaglandins

Table 1: It is not clear to me, what “Positive medical history” means. Please clarify.

Figure 2A, 2B: The lines and text are quite small, and the axis range are a bit to high. Readability is reduced.

Figure 2 appears before Figure 1.

All figures: “A” and “B” should be added to the figures themselves to be unconfusingly identifiable.

Table 2: Not sure what Dellen means (of the cornea?).

Figure 3: consider separating the figures from the table part of the figure.

Line 251: remove “3.1. Subsection”

Line 274: a period is missing at the end of the line.

Author Response

Reviewer: 2

Comments to the Author

Dear authors,

Congratulations on successfully conducting this relevant and well-designed study. I only have a few comments/ questions about your paper.

Major comments:

Could you make a statement about the glaucoma etiology of the patients included in the study (primary/ secondary glaucoma, exfoliation, etc.), as this is important to get an idea about the severity/ aggressivity of the disease.

REPLY and Change: Most studies on the performance of the PRESERFLO-implant (retrieved from the systematic review) included a mixture of different types of glaucoma. Therefore, we did not mention this on the current study population: the majority had POAG, ~16% had PACG, and a similar amount had secondary glaucoma (due to uveitis, pseudo-exfoliation, or steroids). We have added this information to the Results section.

Regarding the meta-analysis, would it be possible to also analyze PRESERFLO with vs. without combined phacoemulsification?

REPLY and Changes: According to Table S1 there are only three small studies that included combined-procedures; however, with the data that these researchers provided in their papers, it is very hard to perform a meaningful meta-analysis.

Minor comments:

Methods:

It seems like only one eye per patient was analyzed, but it is not clearly stated in the methods section. It would be important to clarify this.

REPLY and Changes: This is not correct. We have clarified this in the Methods and Results section.

Is it true, that you didn’t use any antimetabolite for the bleb revisions?

REPLY: That is correct. Mitomycin-C was only used in primary surgery.

Results:

Line 106: typo in prostaglandins

REPLY and Changes: Thanks! This is has now been corrected.

Table 1: It is not clear to me, what “Positive medical history” means. Please clarify.

REPLY and Changes: See reply to reviewer 1. We have removed this row from the Table.

Figure 2A, 2B: The lines and text are quite small, and the axis range are a bit to high. Readability is reduced.

REPLY: This has been done by editing process by the MDPI website. Maybe they can enlarge the figure?

Figure 2 appears before Figure 1.

REPLY: Maybe MDPI can sort the Figures?

All figures: “A” and “B” should be added to the figures themselves to be unconfusingly identifiable.

REPLY: We agree; however, MDPI removed these letters from the Figures during the editing process. Maybe they can put these letters back to the Figures.

Table 2: Not sure what Dellen means (of the cornea?).

REPLY and Changes: Exactly, we have clarified this in the Table 2.

Figure 3: consider separating the figures from the table part of the figure.

Line 251: remove “3.1. Subsection”

REPLY and Changes: This has probably also to do with the editing by MDPI. We have removed the requested part.

Line 274: a period is missing at the end of the line.

REPLY and Changes: Done.

Reviewer 3 Report

I have only minor comments.

1. Why did visual acuity improve after PRESERFLO MicroShunt as cataract-combined procedure?

2. Lines 263-264 At the end of follow-up the mean IOP was almost similar (15.96 vs. 110.7 mmHg). I do not think it is similar.

Author Response

Reviewer: 3

Comments to the Author

I have only minor comments.

  1. Why did visual acuity improve after PRESERFLO MicroShunt as cataract-combined procedure?

REPLY: This is most probably because of the cataract that was removed and replaced for a clear intraocular lens in the combined procedures.

  1. Lines 263-264 At the end of follow-up the mean IOP was almost similar (15.96 vs. 110.7 mmHg). I do not think it is similar.

REPLY and Change: We agree that it is only the IOP-lowering medication that was similar. We have removed this sentence.